# Improving Identically Distributed and Out-of-Distribution Medical Image Classification with Segmentation-Guided Attention in Small Dataset Scenarios

**Mariia Rizhko**[1]                                    MARIIA.RIZHKO@MAIL.UTORONTO.CA
[1] *Department of Computer Science, University of Toronto*

**Lauren Erdman**[1,2]                                  LARUNERDMAN1@GMAIL.COM
[2] *Center for Computational Medicine, Hospital for Sick Children*

**Mandy Rickard**[3]                                    MANDY.RICKARD@SICKKIDS.CA
**Armando J. Lorenzo**[3,4]                             ARMANDO.LORENZO@SICKKIDS.CA
[3] *Division of Urology, Hospital for Sick Children*
[4] *Department of Surgery, University of Toronto*

**Kunj Sheth**[5]                                       KUNJ.SHETH@GMAIL.COM
**Daniel Alvarez**[5]                                   ALVAREZD93@GMAIL.COM
**Kyla N Velaer**[5]                                    KYLA.VELAER@GMAIL.COM
[5] *Stanford Children's Health, Lucile Packard Children's Hospital, Stanford University*

**Megan A. Bonnett**[6]                                 BONNME01@LUTHER.EDU
**Christopher S. Cooper**[6]                            CHRISTOPHER-COOPER@UIOWA.EDU
[6] *Department of Urology, University of Iowa*

**Gregory E. Tasian**[7,8]                              TASIANG@CHOP.EDU
**John Weaver**[7,8]                                    JWEAVE2925@GMAIL.COM
**Alice Xiang**[7,8]                                    ALICE.XIANG@JEFFERSON.EDU
[7] *Department of Surgery, University of Pennsylvania*
[8] *Division of Urology, Children's Hospital of Philadelphia*

**Anna Goldenberg**[1,9]                                NYULIK@GMAIL.COM
[9] *Genetics and Genome Biology, Hospital for Sick Children*

**Editors:** Accepted for publication at MIDL 2024

## Abstract

We propose a new approach for training medical image classification models using segmentation masks, particularly effective in small dataset scenarios. By guiding the model's attention with segmentation masks toward relevant features, we significantly improve accuracy for diagnosing Hydronephrosis. Evaluation of our model on identically distributed data showed either the same or better performance with improvement up to 0.28 in AUROC and up to 0.33 in AUPRC. Our method showed better generalization ability than baselines, improving from 0.02 to 0.75 in AUROC and from 0.09 to 0.47 in AUPRC for four different out-of-distribution datasets. The results show that models trained on smaller datasets using our approach can achieve comparable results to those trained on datasets 25 times larger. The source code is available at github.com/MeriDK/segmentation-guided-attention.

**Keywords:** Medical Image Classification, Domain Generalization, Hydronephrosis.

## 1. Introduction

Can a model trained to predict medical diagnoses for young children maintain the same level of accuracy for older children? If a model is trained with data from one hospital, will it perform just as efficiently with data from another? How precise will a model be when analyzing images produced by different machines? Machine Learning (ML) models struggle with these scenarios since they are trained on *identically distributed* (i.i.d.) data. However, their accuracy can vary significantly when these models are tested on *out-of-distribution* (OOD) data. The problem is known as *domain shift* between i.i.d *source* and OOD *target* domains. It occurs when models are not trained to deal with the domain shift in mind. This issue is significant for the medical field, where labeled data is limited, and training different models for each scenario is impractical.

Domain shift is a challenge that extends beyond healthcare. The task of addressing domain shift is known as *Domain Generalization* (DG), a problem that exists in almost every application of ML (Zhou et al., 2022). For example, in the semantic segmentation task in autonomous driving, a model trained on urban data may fail in rural settings (Hoffman et al., 2018; Ros et al., 2016), potentially leading to accidents. In personal identification systems, the model trained on well-illuminated images may not recognize a person in dim lighting (Sun et al., 2019; Li et al., 2020), potentially preventing access to their home if the lights are broken. Even with seemingly simple tasks like recognizing handwritten digits, ML models can underperform due to minor variations like ink color (Ganin and Lempitsky, 2015). Similarly, in the medical domain, a model trained on images collected with one protocol might be ineffective for images collected through another (Liu et al., 2020). These examples underline the significance of the problem across different domains.

An excellent survey (Zhou et al., 2022) categorizes various DG methodologies. The *Domain Alignment* (Li et al., 2018b,d) methods focus on learning a mapping function between the domain and target datasets. *Meta-learning* (Li et al., 2018a; Balaji et al., 2018) approaches divide data into meta-train and meta-test sets, where a model is trained on the meta-train set and evaluated on the meta-test set. The methods separate domain-specific and domain-agnostic features within datasets in the *Learning Disentangled Representations* (Li et al., 2017; Ilse et al., 2020) category. While Domain Alignment, Meta-learning, and Learning Disentangled Representations offer promising approaches, they require a labeled target dataset during training on a domain dataset. The target dataset is usually unavailable during training in the medical domain, so other approaches should be used.

The DG survey (Zhou et al., 2022) also covers the methodologies that do not require a target dataset during the training. *Data augmentations* (Volpi et al., 2018; Volpi and Murino, 2019; Xu et al., 2020) simulate a domain shift by changing images. *Ensemble learning* (Xu et al., 2014; Cha et al., 2021) trains the same model with a different random seed for weight initialization or data split. *Self-supervised learning* (Carlucci et al., 2019; Bucci et al., 2021) lets a model learn generic features of your data first and then fine-tune the model for a downstream task. *Regularization Strategies* (Wang et al., 2019; Huang et al., 2020) learn generalized features by focusing on global structure instead of local patterns or by masking out over dominant features. All of these approaches are generally considered to make more robust models. However, when trained on small datasets, which is usually the case for the medical domain, their performance might suffer on i.i.d and OOD data.

before    after    before    after

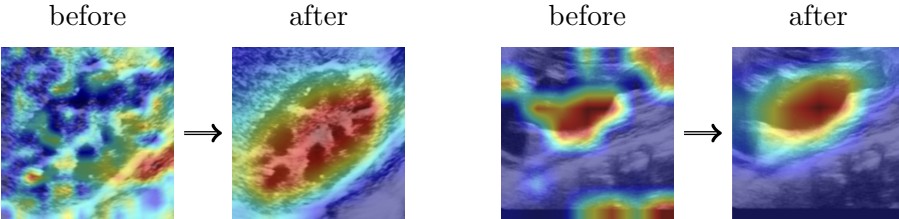

Figure 1: Illustration of our approach. Before: focus on both organ and background noise. After: targeted focus on the critical organ.

To address this limitation we utilize Gradient-weighted Class Activation Mapping (Grad-CAM) (Selvaraju et al., 2017) to create attention maps. GradCAM is a visual explanation method in computer vision that highlights the regions in an input image that influence the model's outcome the most. Prior work showed that this attention mechanism could be learned, resulting in a better performance for image segmentation (Li et al., 2018c) and classification tasks (Fukui et al., 2019). This idea has been adapted for the medical domain: it showed improved accuracy for thyroid nodules (Lu et al., 2022), for chest X-ray abnormality localization and diagnosis (Ouyang et al., 2020b), for diagnosis of COVID-19 (Ouyang et al., 2020a) and dementia (Lian et al., 2020, 2019). However, these prior studies used large datasets, ranging from 2,000 MRI scans to as many as 1.2 million images from ImageNet, and they focused only on i.i.d. data. In contrast, we apply the idea to DG tasks and demonstrate its effectiveness on small datasets with less than 100 images.

Our study addresses the typical scenario in the medical field where models are trained on small datasets. Typically, these models learn specific "useful" noise patterns, leading to high performance on similar (i.i.d) test datasets. However, their accuracy declines when applied to new images without these noise patterns. Beyond basic classification tasks, models should also be trained to disregard features known a priori to be irrelevant. For instance, in kidney ultrasound classification (see Figure 1), the model should focus only on the kidney, ignoring background noise. Often, in medical imaging, additional information like segmentation masks is available. We adapt gradient-based techniques to utilize segmentation masks for medical imaging with small dataset scenarios. This adaptation allows us to effectively train models on small datasets and improve their performance when tested on i.i.d and OOD data without having target datasets during training.

## 2. Method

Let $\mathcal{X}$ be the input image space and $\mathcal{Y}$ the label space. A *domain* is defined as a joint distribution $\mathcal{D} = (\mathcal{X}, \mathcal{Y})$, which contains image-label pairs $\{(x^{(n)}, y^{(n)})\}_{n=1}^{N}$, where N is the number of samples. Our goal is to learn a classification model $F_\theta : \mathcal{X} \to \mathcal{Y}$ using the source domain $\mathcal{D}$ for generalization across unseen target domains $\{\mathcal{D}_{tg}^1, \mathcal{D}_{tg}^2, \ldots, \mathcal{D}_{tg}^K\}$ set of K target domains. In a source domain for input images $\mathcal{X}$, we have corresponding segmentation masks $\mathcal{M}$ that will be utilized for our method. Note that there are no requirements for masks in target domains. The core idea of our method is to force the model to learn two

things simultaneously: the attention mechanism learning task and the classification task itself.

**Attention Map Calculation.** Given an input image, a classification model processes it up to a target layer. Let $A^k$ represent the activation of the $k$-th feature map at this layer. The gradient of the score for class $c$, denoted $y^c$, with respect to the activations $A^k$ of the feature map is computed. This gradient is represented as $\frac{\partial y^c}{\partial A^k}$. To obtain the neuron importance weights $\alpha_k^c$ we apply Global Average Pooling (GAP) to these gradients. This is given by:

$$\alpha_k^c = GAP\left(\frac{\partial y^c}{\partial A^k}\right) \tag{1}$$

The Class Activation Map (CAM) for class $c$, denoted as $L_{Grad-CAM}^c$, is a weighted sum of the feature maps, weighted by $\alpha_k^c$, and passed through a ReLU function:

$$L_{Grad-CAM}^c = ReLU\left(\sum_k \alpha_k^c A^k\right) \tag{2}$$

The final Attention Map $\mathcal{A}$ is achieved by resizing $L_{Grad-CAM}^c$ to the dimensions of the input image.

**Attention Loss.** It is a custom loss function, denoted as $\mathcal{L}_{Attention}$, that incorporates the difference between the Grad-CAM Attention Map $\mathcal{A}$ and a given ground truth attention mask $\mathcal{M}$ by calculating the mean squared error (MSE) between $\mathcal{A}$ and $\mathcal{M}$:

$$\mathcal{L}_{Attention} = \frac{1}{N}\sum_{i=1}^{N}(\mathcal{A}_i - \mathcal{M}_i)^2 \tag{3}$$

where $N$ is the total number of pixels in the image, and $i$ indexes these pixels.

This loss function measures the alignment between the regions highlighted by the Grad-CAM and those indicated by the attention mask. The objective of the training is to minimize this loss, thereby encouraging the model to focus more on areas marked as important by the mask.

**Overall Loss.** The Binary Cross-Entropy Loss $\mathcal{L}_{BCE}$, given the predicted outputs $y^{pred}$ and the true labels $y^{true}$, is defined as:

$$\mathcal{L}_{BCE} = -\frac{1}{N}\sum_{i=1}^{N}\left[y_i^{true}\log(y_i^{pred}) + (1 - y_i^{true})\log(1 - y_i^{pred})\right] \tag{4}$$

where $N$ is the number of samples and $i$ indexes these samples.

The overall loss is a weighted combination of the Binary Cross-Entropy Loss $\mathcal{L}_{BCE}$ and the Attention Loss $\mathcal{L}_{Attention}$.

$$\mathcal{L} = \alpha\mathcal{L}_{BCE} + \beta\mathcal{L}_{Attention} \tag{5}$$

where $\alpha$ and $\beta$ are weighting coefficients that balance the two components of the loss. In all our experiments $\alpha = \beta = 1$.

By combining these two losses, the model not only focuses on minimizing the prediction error but also emphasizes the alignment of the attention maps with the important regions as marked by the attention masks.

## 3. Experiments

Hydronephrosis (HN) is a medical condition characterized by the swelling of one or both kidneys due to a urine buildup. It can affect people of any age and is spotted in up to 5% of babies during routine pregnancy ultrasound scans. However, surgical intervention is only required in 20% of these cases (Dos Santos et al., 2015). To determine which cases need intervention, patients receive repeated invasive scans and ultrasounds to monitor whether the HN is causing functional damage or resolving without the need for surgery. Previously, deep learning models have used postnatal ultrasound images to predict surgical intervention in HN from the first ultrasound (Erdman et al., 2020), further investigated predicting HN grades (Smail et al., 2020) and risk scores (Tabrizi et al., 2021). While prior models (Erdman et al., 2020; Smail et al., 2020; Tabrizi et al., 2021) worked well for i.i.d data, they showed lower performance on smaller and OOD datasets.

**Datasets.** We use five datasets from four pediatric hospitals in North America containing ultrasounds of kidneys. The variation of the data comes not only from its collection across various hospitals but also from differences in patient demographics and imaging equipment, attributing to its OOD characteristics. For example, the average patient age in the Hospital for Sick Children (SickKids) is 53 weeks, while in the Children's Hospital of Philadelphia (CHOP), it is 313 weeks. These variances provide a robust setting for evaluating models for DG.

The source domain dataset $\mathcal{D}$ from SickKids has 2542 ultrasounds. 20% of $\mathcal{D}$ is held out to create i.i.d. test dataset $\mathcal{D}_{test}$. The rest of the 2048 images are used for training and validation. Only 83 out of 2048 images have corresponding kidney segmentation masks. We will utilize these 83 images for training baseline models. The same 83 images with their segmentation masks will also be used to train our model. To assess the robustness of our model, we will further use the complete set of 2048 images, which is bigger by 25 times, to train additional baseline models. We split $\mathcal{D}$ into train $\mathcal{D}_{train}$ and validation $\mathcal{D}_{val}$ sets, ensuring each patient's images appear in only one set. $\mathcal{D}_{train}$ has only 66 images with kidney segmentation masks, creating $\mathcal{D}_{train}^{seg}$ with 51 non-surgical and 15 surgical cases. Similarly, the validation set $\mathcal{D}_{val}^{seg}$ is a subset of $\mathcal{D}_{val}$ and has 10 non-surgical and 7 surgical cases.

We evaluate models on four distinct target domain datasets. The first, $T_{SickKids}$, includes data from 202 patients at SickKids, having 711 images, of which 75 are positive. Despite being collected at the same hospital as the training dataset, patient demographics and imaging equipment variations make this dataset OOD. The second dataset, $T_{Stanford}$, is from the Stanford Children's Hospital (Stanford) and includes data from 103 patients, with 551 images (27 positive). The third, $T_{UIowa}$, is from the University of Iowa Children's Hospital (UIowa) with 91 patients and 97 images (56 positives). Lastly, $T_{CHOP}$ comes from CHOP with 89 patients and 89 images, 60 of which are positive. The datasets summary is shown in Table 1.

**Baselines.** ResNet-18, ResNet-50 (He et al., 2016), ViT-Tiny, and ViT-Base (Dosovitskiy et al., 2020) were trained on $\mathcal{D}_{train}^{seg}$ and validated on $\mathcal{D}_{val}^{seg}$, utilizing Binary Cross Entropy Loss $\mathcal{L}_{BCE}$ only. We tested two weights initialization methods: Kaiming uniform initialization (random) (He et al., 2015) and using weights pre-trained on ImageNet (Deng et al., 2009). Hyperparameters were tuned via Bayesian optimization (Snoek et al., 2012) to minimize the loss on $\mathcal{D}_{val}^{seg}$, more details in Appendix A. Consistent image transformations

Table 1: Datasets Summary

| Name | Hospital | Domain | Used for | Masks | Patients | Images | Pos | Neg |
|---|---|---|---|---|---|---|---|---|
| $D_{train}$ | SickKids | source | training | $\times$ | 266 | 1549 | 185 | 1364 |
| $D_{train}^{seg}$ | SickKids | source | training | $\checkmark$ | 35 | 66 | 15 | 51 |
| $D_{val}$ | SickKids | source | validation | $\times$ | 89 | 499 | 67 | 432 |
| $D_{val}^{seg}$ | SickKids | source | validation | $\checkmark$ | 7 | 17 | 7 | 10 |
| $D_{test}$ | SickKids | source | i.i.d. test | $\times$ | 89 | 494 | 71 | 423 |
| $T_{SickKids}$ | SickKids | target | OOD test | $\times$ | 202 | 711 | 75 | 636 |
| $T_{Stanford}$ | Stanford | target | OOD test | $\times$ | 103 | 551 | 27 | 524 |
| $T_{UIowa}$ | UIowa | target | OOD test | $\times$ | 91 | 97 | 56 | 41 |
| $T_{CHOP}$ | CHOP | target | OOD test | $\times$ | 89 | 89 | 60 | 29 |

are applied across all experiments. Rotation, cropping, horizontal flipping, and normalization are used for training, while resizing and normalization are used for validation. We train all models for 30 epochs with early stopping based on validation AUROC. One NVIDIA RTX 2080 Ti was used for all experiments. For further analysis, we also trained additional baseline models on the larger datasets $\mathcal{D}_{train}$ and $\mathcal{D}_{val}$; all experimental setups were the same.

**Our model.** We trained the ResNet-18 model on $\mathcal{D}_{train}^{seg}$ and validated it on $\mathcal{D}_{val}^{seg}$, utilizing our proposed loss function as described in Equation 5. The model's weights were initialized using pre-trained ImageNet weights. All other experimental setups, including hyperparameters search, image transformations, and training duration, were consistent with those used in the baseline models.

### 3.1. Baselines vs. Our Model trained on the Small Datasets.

The results in Table 2 and Table 3 show Area Under the Receiver Operating Characteristic (AUROC) and Area Under the Precision-Recall Curve (AUPRC) of the baseline models and our model, all trained on $\mathcal{D}_{train}^{seg}$ and validated on $\mathcal{D}_{val}^{seg}$.

**I.i.d. Comparison.**

Table 2 presents a comparative analysis of the models performance on i.i.d. test dataset $\mathcal{D}_{test}$. Note, $\mathcal{D}_{test}$ is a held-out test dataset from the whole dataset $\mathcal{D}$ and has 494 images, while the models are trained on the small subsets $\mathcal{D}_{train}^{seg}$ and $\mathcal{D}_{val}^{seg}$ with 66 and 17 images respectively. This comparison reflects each model's ability to generalize to new data with a similar distribution to the training set. Interestingly, only three models, including our own, were able to effectively generalize to $\mathcal{D}_{test}$, showing 0.81-0.83 AUROC and 0.48 AUPRC.

**OOD Comparison.**

Table 3 presents the performance of the models across four different OOD datasets $T_{SickKids}$, $T_{Stanford}$, $T_{UIowa}$, and $T_{CHOP}$. Notably, our model consistently outperformed all baselines across all OOD datasets.

Table 2: Comparison of models trained on the small dataset $\mathcal{D}_{train}^{seg}$ for performance on held-out i.i.d. test dataset $\mathcal{D}_{test}$

| Model Name | Backbone | Weights Init. | AUROC | AUPRC |
|---|---|---|---|---|
| R18-random-small | ResNet-18 | Random | 0.79 | 0.43 |
| R18-imagenet-small | ResNet-18 | ImageNet | 0.70 | 0.27 |
| R50-random-small | ResNet-50 | Random | 0.68 | 0.30 |
| R50-imagenet-small | ResNet-50 | ImageNet | 0.71 | 0.28 |
| ViT-T-random-small | ViT-Tiny | Random | 0.55 | 0.20 |
| ViT-T-imagenet-small | ViT-Tiny | ImageNet | **0.81** | **0.48** |
| ViT-B-random-small | ViT-Base | Random | 0.54 | 0.15 |
| ViT-B-imagenet-small | ViT-Base | ImageNet | **0.83** | **0.48** |
| R18-attention (Ours) | ResNet-18 | ImageNet | **0.82** | **0.48** |

Table 3: Comparison of models trained on the small dataset $\mathcal{D}_{train}^{seg}$ for performance on four different OOD datasets $T_{SickKids}$, $T_{Stanford}$, $T_{UIowa}$, and $T_{CHOP}$

| | AUROC | | | | AUPRC | | | |
|---|---|---|---|---|---|---|---|---|
| Model Name | $T_{SickKids}$ | $T_{Stanford}$ | $T_{UIowa}$ | $T_{CHOP}$ | $T_{SickKids}$ | $T_{Stanford}$ | $T_{UIowa}$ | $T_{CHOP}$ |
| R18-random-small | 0.47 | 0.19 | 0.33 | 0.34 | 0.09 | 0.04 | 0.48 | 0.59 |
| R18-imagenet-small | 0.52 | 0.35 | 0.72 | 0.54 | 0.10 | 0.04 | 0.77 | 0.73 |
| R50-random-small | 0.52 | 0.23 | 0.39 | 0.27 | 0.20 | 0.04 | 0.51 | 0.58 |
| R50-imagenet-small | 0.37 | 0.13 | 0.18 | 0.21 | 0.08 | 0.03 | 0.43 | 0.52 |
| ViT-T-random-small | 0.52 | 0.29 | 0.76 | 0.56 | 0.10 | 0.03 | 0.73 | 0.72 |
| ViT-T-imagenet-small | 0.80 | 0.72 | 0.80 | 0.69 | 0.35 | 0.22 | 0.85 | 0.82 |
| ViT-B-random-small | 0.46 | 0.31 | 0.71 | 0.50 | 0.10 | 0.03 | 0.74 | 0.68 |
| ViT-B-imagenet-small | 0.84 | 0.84 | 0.72 | 0.68 | 0.46 | 0.33 | 0.79 | 0.81 |
| R18-attention (Ours) | **0.86** | **0.88** | **0.90** | **0.81** | **0.53** | **0.42** | **0.90** | **0.92** |

### 3.2. Baselines trained on the Big Datasets vs. Our Model trained on the Small Datasets.

To further analyze our model, we trained additional baselines on the whole train dataset $\mathcal{D}_{train}$ and validation dataset $\mathcal{D}_{val}$ with a total of 2078 images. We compared the baselines to our model trained on $\mathcal{D}_{train}^{seg}$ and validated on $\mathcal{D}_{val}^{seg}$ with a total of 83 images.

**I.i.d. Comparison.**

Table 4 shows the overall performance of baselines and our model on the held-out i.i.d. test dataset $\mathcal{D}_{test}$. All models, including our trained only **on 4% of the data**, have comparable AUROC (0.82 - 0.87) and AUPRC (0.47 - 0.54), which means all models generalize well to unseen images from the same i.i.d. distribution.

**OOD Comparison.**

Table 5 shows models' performance on OOD datasets $T_{SickKids}$, $T_{Stanford}$, $T_{CHOP}$, and $T_{UIowa}$. Out of 9 models that perform well on i.i.d. data, only three models, including ours, transfer well to all OOD datasets. It demonstrates the effectiveness of using our approach, considering that our model trained on 25 times less data could generalize well to OOD data.

Table 4: Comparison of baselines trained on the big dataset $\mathcal{D}_{train}$ and our model trained on the small dataset $\mathcal{D}_{train}^{seg}$ for performance on held-out i.i.d. test dataset $\mathcal{D}_{test}$

| Model Name | Backbone | Weights Init. | Images | AUROC | AUPRC |
|---|---|---|---|---|---|
| R18-random | ResNet-18 | Random | 2078 | 0.85 | 0.50 |
| R18-imagenet | ResNet-18 | ImageNet | 2078 | 0.87 | 0.52 |
| R50-random | ResNet-50 | Random | 2078 | 0.82 | 0.47 |
| R50-imagenet | ResNet-50 | ImageNet | 2078 | 0.83 | 0.52 |
| ViT-T-random | ViT-Tiny | Random | 2078 | 0.84 | 0.50 |
| ViT-T-imagenet | ViT-Tiny | ImageNet | 2078 | 0.86 | 0.54 |
| ViT-B-random | ViT-Base | Random | 2078 | 0.84 | 0.49 |
| ViT-B-imagenet | ViT-Base | ImageNet | 2078 | 0.85 | 0.52 |
| R18-attention (Ours) | ResNet-18 | ImageNet | **83** | 0.82 | 0.48 |

Table 5: Comparison of baselines trained on the big dataset $\mathcal{D}_{train}$ and our model trained on the small dataset $\mathcal{D}_{train}^{seg}$ for performance on four OOD datasets $T_{SickKids}$, $T_{Stanford}$, $T_{UIowa}$, and $T_{CHOP}$

| | AUROC | | | | AUPRC | | | |
|---|---|---|---|---|---|---|---|---|
| Model Name | $T_{SickKids}$ | $T_{Stanford}$ | $T_{UIowa}$ | $T_{CHOP}$ | $T_{SickKids}$ | $T_{Stanford}$ | $T_{UIowa}$ | $T_{CHOP}$ |
| R18-random | 0.59 | 0.35 | 0.43 | 0.36 | 0.19 | 0.04 | 0.52 | 0.59 |
| **R18-imagenet** | **0.88** | **0.88** | **0.82** | **0.85** | **0.55** | **0.40** | **0.88** | **0.94** |
| R50-random | 0.61 | 0.49 | 0.66 | 0.55 | 0.21 | 0.06 | 0.67 | 0.71 |
| R50-imagenet | 0.74 | 0.66 | *0.80* | *0.78* | 0.23 | 0.07 | *0.84* | 0.84 |
| ViT-T-random | 0.53 | 0.17 | 0.23 | 0.27 | 0.17 | 0.03 | 0.43 | 0.55 |
| ViT-T-imagenet | 0.77 | 0.62 | 0.66 | 0.62 | 0.35 | 0.12 | 0.72 | 0.72 |
| ViT-B-random | 0.57 | 0.22 | 0.23 | 0.24 | 0.24 | 0.04 | 0.44 | 0.54 |
| **ViT-B-imagenet** | **0.89** | **0.91** | **0.88** | **0.85** | **0.55** | **0.48** | **0.88** | **0.93** |
| **R18-attention (Ours)** | **0.86** | **0.88** | **0.90** | **0.81** | **0.53** | **0.42** | **0.90** | **0.92** |

## 4. Conclusion

This paper presented a new method for improving medical image classification models using segmentation masks, especially effective in small dataset scenarios (less than 100 images). By utilizing a specialized loss function, our model demonstrated remarkable performance on both i.i.d. and OOD datasets despite limited training data. It matched or exceeded the performance of other models trained on similar-sized datasets in i.i.d. scenarios and consistently outperformed all baselines in OOD settings. Notably, our model, trained on just 4% of the data, showed the same or even better performance as baselines trained on significantly larger datasets in i.i.d. and OOD settings. The implications of these results are promising. Creating segmentation masks, which our method relies on, could be more feasible than gathering extensive data on rare diseases. Additionally, our model's ability to transfer across different hospitals could reduce the need for unique models for each medical setting.

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

## Appendix A. Hyperparameterms Search

In our study, we used Bayesian optimization to systematically explore and identify optimal hyperparameters for training all models. We focused on tuning batch size, gamma, learning rate, and weight decay, aiming to minimize the validation loss. We tested batch sizes of 16, 32, 64, and 128; gamma values ranging from 0.99 to 0.85 in decrements of 0.02; learning rates of 0.1, 0.01, 0.001, 0.0001, 1e-05, and 1e-06; and weight decay parameters of 0.3, 0.1, 0.03, 0.01, 0.003, and 0.001. The best set of hyperparameters for each model is reported in the Table 6.

Table 6: Hyperparameter Selection for Models

| Model Name | Batch Size | Gamma | Learning Rate | Weight Decay |
|---|---|---|---|---|
| R18-random-small | 32 | 0.91 | 0.001 | 0.1 |
| R18-imagenet-small | 16 | 0.91 | 0.001 | 0.1 |
| R50-random-small | 32 | 0.85 | 0.01 | 0.01 |
| R50-imagenet-small | 64 | 0.85 | 0.01 | 0.01 |
| ViT-T-random-small | 64 | 0.85 | 0.01 | 0.003 |
| ViT-T-imagenet-small | 128 | 0.95 | 0.0001 | 0.01 |
| ViT-B-random-small | 16 | 0.85 | 0.000001 | 0.3 |
| ViT-B-imagenet-small | 128 | 0.91 | 0.00001 | 0.001 |
| R18-random | 64 | 0.89 | 0.001 | 0.001 |
| R18-imagenet | 64 | 0.93 | 0.00001 | 0.01 |
| R50-random | 64 | 0.87 | 0.001 | 0.001 |
| R50-imagenet | 16 | 0.93 | 0.0001 | 0.001 |
| ViT-T-random | 32 | 0.87 | 0.0001 | 0.001 |
| ViT-T-imagenet | 16 | 0.99 | 0.00001 | 0.01 |
| ViT-B-random | 16 | 0.91 | 0.00001 | 0.001 |
| ViT-B-imagenet | 16 | 0.87 | 0.000001 | 0.03 |
| R18-attention (Ours) | 128 | 0.85 | 0.001 | 0.1 |

## Appendix B. Attention Score

To quantify how much different models actually pay attention to the region of interest, we create a new metric *Attention Score*, which has two components Overlap Score and Coverage Score. *Overlap Score (OS)* measures the proportion of the important areas, as defined by the ground truth mask $\mathcal{M}$, that is successfully captured by the attention map $\mathcal{A}$:

$$OS(\mathcal{A}, \mathcal{M}) = \frac{\sum_{i=1}^{N} \min(\mathcal{A}_i, \mathcal{M}_i)}{\sum_{i=1}^{N} \mathcal{M}_i} \tag{6}$$

where $N$ is the total number of pixels, and $i$ indexes these pixels. *Coverage Score (CS)* assesses the concentration and specificity of the model's attention, evaluating how much of the attention map's activation $\mathcal{A}$ is meaningfully focused on the target areas $\mathcal{M}$:

$$CS(\mathcal{A}, \mathcal{M}) = \frac{\sum_{i=1}^{N} \min(\mathcal{A}_i, \mathcal{M}_i)}{\sum_{i=1}^{N} \mathcal{A}_i} \tag{7}$$

where $N$ is the total number of pixels, and $i$ indexes these pixels. You can think about the Overlap Score as a Recall metric and the Coverage Score as Precision but for Attention maps instead of classification labels. The final Attention Score is computed as the harmonic mean of the Overlap Score and Coverage Score, providing a balanced measure of both overlap and coverage:

$$AttentionScore(\mathcal{A}, \mathcal{M}) = 2 \times \frac{OS(\mathcal{A}, \mathcal{M}) \times CS(\mathcal{A}, \mathcal{M})}{OS(\mathcal{A}, \mathcal{M}) + CS(\mathcal{A}, \mathcal{M})} \tag{8}$$

Table 7: Attention Scores on i.i.d. $\mathcal{D}_{test}$ and OOD datasets $T_{SickKids}$, $T_{Stanford}$, $T_{UIowa}$, and $T_{CHOP}$

| Model Name | $\mathcal{D}_{test}$ | $T_{SickKids}$ | $T_{Stanford}$ | $T_{UIowa}$ | $T_{CHOP}$ |
|---|---|---|---|---|---|
| R18-random | 0.38 | 0.51 | 0.34 | 0.46 | 0.44 |
| R18-imagenet | 0.43 | 0.56 | 0.52 | 0.57 | 0.52 |
| R50-random | 0.40 | 0.49 | 0.40 | 0.40 | 0.51 |
| R50-imagenet | 0.41 | 0.48 | 0.43 | 0.45 | 0.45 |
| ViT-T-random | 0.44 | 0.57 | 0.10 | 0.23 | 0.09 |
| ViT-T-imagenet | 0.26 | 0.41 | 0.32 | 0.43 | 0.42 |
| ViT-B-random | 0.45 | 0.59 | 0.50 | **0.67** | 0.58 |
| ViT-B-imagenet | 0.27 | 0.36 | 0.38 | 0.39 | 0.39 |
| R18-attention (Ours) | **0.57** | **0.60** | **0.61** | 0.59 | **0.62** |

In Table 7, we show that our model generally outperforms other models in terms of Attention Score. The interesting exception is the Attention Score for the ViT-B-random model on $T_{UIowa}$ dataset, where it shows a higher score than our model. Considering the low performance of the ViT-B-random model in terms of AUROC and AUPRC on that dataset, we conclude that the attention score, even though it is a useful indicator of the model's performance, is only a part of the evaluation and should be considered in combination with other metrics.

## Appendix C. Datasets Information

**Training Dataset**
*Sex distribution:* 2027 M, 515 F.
*Kidney side distribution:* 1289 Left, 1253 Right.
*Ultrasound machine distribution:* philips-medical-systems: 992, toshiba-mec: 497, NA: 376, ToshibaST: 258, PhilipsST: 112, SamsungST: 97, ge-medical-systems: 45, samsung-medison-co-ltd: 36, OutsideST: 26, acuson: 25, atl: 22, toshiba-mec-us: 20, TreeST: 17, GEST: 13, siemens: 4, ge-healthcare: 2.
*The age* varies from 0.14 weeks to 720 weeks, with an average of 53 weeks.

**OOD dataset** $T_{SickKids}$
*Sex distribution:* 599 M, 112 F.
*Kidney side distribution:* 475 Left, 236 Right.

*Ultrasound machine distribution:* ToshibaST: 294, PhilipsST: 247, SamsungST: 158, OutsideST: 12.

*The age* varies from 0.29 weeks to 92 weeks, with an average of 17 weeks.

**OOD dataset** $T_{Stanford}$

*Sex distribution:* 413 M, 138 F.

*Kidney side distribution:* 275 Left, 276 Right.

*Ultrasound machine distribution:* Stanford: 551.

*The age* varies from 104.0 weeks to 988 weeks, with an average of 190 weeks.

**OOD dataset** $T_{UIowa}$

*Sex distribution:* 80 M, 17 F.

*Kidney side distribution:* 59 Left, 38 Right.

*Ultrasound machine distribution:* UIowa: 97.

*The age* varies from 0.14 weeks to 266 weeks, with an average of 28 weeks.

**OOD dataset** $T_{CHOP}$

*Sex distribution:* 55 M, 34 F.

*Kidney side distribution:* 56 Left, 33 Right.

*Ultrasound machine distribution:* Philips: 51, GE: 16, Phillips: 7, HDI 5000: 3, Siemens: 2, Acuson: 2, General electric: 1, MRI abd w/wo, RBUS 7/19/2010: 1, Cineloop: 1, Mindray: 1, Toshiba: 1.

*The age* varies from 1.43 weeks to 1001 weeks, with an average of 313 weeks.

## Appendix D. Limitations and Future Work

**Limitations.** Even though our model performed well on multiple out-of-distribution datasets, it's worth noting that all the data came from hospitals in the USA and Canada. In real-world scenarios, particularly in areas with substantially different demographics or medical equipment, our model might show diminished performance.

**Future Work.** We plan to conduct a series of comprehensive ablation studies to precisely quantify the impact of attention loss on the performance of each model separately. Additionally, we aim to broaden the applicability and robustness of our model by collecting and incorporating data from hospitals outside North America. This effort will test the model's ability to generalize across diverse demographics, addressing potential biases and enhancing its global applicability. Furthermore, we intend to explore the potential of our approach in other clinical settings, such as the diagnosis of pneumonia in lung ultrasound images. By extending our domain generalization efforts to various medical imaging tasks, we hope to contribute further to the advancement of AI in healthcare, ensuring models are both effective and equitable across different populations and conditions.

