# OpenReview forum: "Improving Identically Distributed and Out-of-Distribution Medical Image Classification with Segmentation-Guided Attention in Small Dataset Scenarios"
_MIDL.io/2024/Conference — MIDL 2024 Poster_

### Official Review · Reviewer_xY5m · 2024-02-22

**Confidence:** 4
**Preliminary Rating:** 4
**Recommendation:** Poster
**Final Rating:** 4

**Summary:**

In this paper, the authors propose to employ attention loss based on Grad-CAM to penalize the model for focusing on undesired image regions. Those regions are defined by segmentation masks. The authors note that this has already been done to improve model performance, but has not been shown on small datasets and OOD samples. They compare this approach with other models trained using standard binary cross entropy as baselines.

**Strengths:**

* Domain generalization and small datasets are both very important issues in medical image analysis
* Systematic evaluation of performance improvements on a) small datasets and b) OOD samples has been made
* Datasets and evaluation metrics are suitable
* Results are critically reflected and contextualized in the limitations-section

**Weaknesses:**

* It could be more clear, which findings (in terms of model performance improvement) have already been made by others and which findings are contributed here. Some important prior work is given in the methods section instead of the introduction
* The paper focuses on both sample efficiency and domain generalization but the introduction only focuses on the latter, while the title only names the former
* The Grad-CAM visualizations in Figure 2 only show the attention of the proposed approach. Without a comparison to the baseline models, this is of limited use. Figure 1 shows such a comparison. It is not clear why those figures are separate.
* The naming of the many different data sets is confusing
* Using multiple cohorts / domains for training is a straightforward way to improve domain generalization. This should have been compared here. How much would that alone improve performance? And how much could attention loss then add to that performance

**Detailed Comments:**

see weaknesses

**Justification Of Final Rating:**

The paper show an interesting validation study that is useful for the community. It applies not only to ultrasound images but could extend to many other medical image analysis tasks. However, the paper is a bit hard to follow especially due to confusing data set naming. Also, the line between prior work and own contribution could be clearer. Also, one quite obvious experiment is missing

**Justification Of The Preliminary Rating:**

The paper show an interesting validation study that is useful for the community. It applies not only to ultrasound images but could extend to many other medical image analysis tasks. However, the paper is a bit hard to follow especially due to confusing data set naming. Also, the line between prior work and own contribution could be clearer. Also, one quite obvious experiment is missing

**Questions To Address In The Rebuttal:**

see weaknesses

**Special Issue:**

No

---

> ### Author Response · Authors · 2024-03-18
>
> Thank you for your time and feedback! We have revised our manuscript to address your concerns as follows:
>
> **Prior Work:** Updated the introduction section so it contains the important prior work instead of the methods section.
>
> **Title Revision:** Thank you for the suggestion! We have updated the title to “Improving Identically Distributed and Out-of-Distribution Medical Image Classification with Segmentation-Guided Attention in Small Dataset Scenarios“ so it more accurately represents the scope of our work.
>
> **Grad-CAM Figures and Datasets Namings:** Updated the figures and standardized dataset names for clarity.
>
> **Why don’t we use multiple domains for training?** We assume that only the source dataset is available during the training. So there will be no need to collect data for each hospital separately but rather just apply the already trained model from another hospital.
>
> We appreciate your insights, which have strengthened our manuscript.

---

> > ### Comment · Reviewer_xY5m · 2024-03-27
> >
> > Thank you for your revision. I understand the rationale behind your answer to my last point regarding using multiple domains for training. However, you are proposing a method that improves domain generalization. One other way of improving domain generalization is using multiple domains during training. Of course, often times this data is not available. But you do have such data and could easily perform a comparison of these 2 approaches and assess how far your approach would lead to domain generalization compared to collecting multiple domains to train on.

---

### Official Review · Reviewer_HeNo · 2024-02-25

**Confidence:** 3
**Preliminary Rating:** 2
**Final Rating:** 2.5

**Summary:**

This paper presents an approach for training medical image classification models using segmentation masks and small dataset. The evaluation focused on improving the accuracy for diagnosing hydronephrosis. The major outcomes of this work is that the results show that models trained on small dataset can achieve comparable results to models based transfer learning trained on datasets 25 times larger.
This paper investigates a significant issue arising when there is a limited dataset available for constructing ML/DL models.

**Strengths:**

The work presented in this paper attempted to address a challenge that extends beyond medical imaging and could be useful for other ML/DL applications. Five datasets were used to test the validity of the proposed approach.

**Weaknesses:**

•	The novelty in this work is limited.

•	The literature review section lacks depth.

•	The idea of training classification models to ignore features of the ROI surrounding areas in medical applications is not convincing. Substantiating such a claim requires input from subject specialists, such as medical experts. It might be valid for some medical applications. However, in several applications, e.g. thyroid nodule classification in US, the identification of the nodule characteristics such as echogenicity requires information from the surrounding area of the nodule.

•	This work assumes additional information like segmentation masks is available. This is not the case in several medical images. In addition the identifications of the ROI boundary in US imaging is highly subjective.

•	The rational of selecting Grad-CAM is not clear. The are several enhancements to the Grad-CAM method, including Ablation-CAM [1] and Clustered-CAM [2]. These methods have demonstrated to be more accurate in US image classification task [4].

•	Several details about the datasets (for example the distributions of the image sizes, imaging equipment or machine maker, ..) are not presented in the paper.

•	51 non-surgical and 15 surgical cases were used. There is no discussion about the impact of class imbalance on the proposed model.

•	The method of producing kidney segmentation masks is not described. It is not clear if the segmentation mask includes the whole boundary of the organ or just the inner boundary.

•	The rational of selecting the baseline networks is not clear.

•	The paper does not present any future work.

[1]. "Ablation-cam: Visual explanations for deep convolutional network via gradient-free localization." proceedings of the IEEE/CVF winter conference on applications of computer vision. 2020.

[2]. "Clustered-CAM: Visual Explanations for Deep Convolutional Networks for Thyroid Nodule Ultrasound Image Classification." In Medical Imaging with Deep Learning. 2021.

[3]. Classification of breast lesions in ultrasound images using deep convolutional neural networks: transfer learning versus automatic architecture design. Medical & Biological Engineering & Computing. 2024.

**Detailed Comments:**

See the comments above.

**Justification Of Final Rating:**

I would like to thank the authors for their efforts during the rebuttal process. They have addressed some of concerns during the rebuttal process. However, the dependency on segmentation masks limitation is not fully addressed. Following the stated argument that creating segmentation masks is relatively straightforward, I would recommend adding the automatic segmentation to the proposed approach and test the method with manual/ automatic segmentations. In addition, the attention loss relies on the attention map from Grad-CAM. Different visualization methods produce different activation maps; how this impact on the proposed method is not discussed in the paper.

**Justification Of The Preliminary Rating:**

See detailed comments in the weaknesses section. Addressing the points in the 'Questions To Address In The Rebuttal' will improve the quality of the paper and evidence the validity of the proposed approach.

**Questions To Address In The Rebuttal:**

•	Provide the key contributions in the introduction section.
•	Comparison with other visualization methods (such as Ablation-cam and/or Clustered-CAM) will improve the quality of the paper.
•	I would like to see a discussion about the impact of class imbalance on the proposed model.
•	I would really like to see more discussion, medical views, and definition of ignoring what so called "background noise".
•	The paper could benefit from a more in-depth literature review.
•	I would like to see more about the future directions of this research.

---

> ### Author Response · Authors · 2024-03-18
>
> Thank you for your constructive feedback! Below, we address the key points raised in your review:
>
> **Focus on Surrounding Areas:** We acknowledge your concern regarding the relevance of surrounding areas in various medical applications. In our experiments, we focus on Hydronephrosis which is the swelling of kidneys, and all the signs of Hydronephrosis are located inside the kidney. When transferring this approach to other medical conditions one should take a medical opinion on what is considered important in the ultrasound.
>
> **Dependency on Segmentation Masks:** This is true that we need extra segmentation masks to train the model. However, in specific medical conditions such as Hydronephrosis, creating segmentation masks for kidneys is relatively straightforward and cost-effective compared to the significant effort and resources needed to collect larger datasets. Our approach leverages these masks to train models that can be easily adapted and reused across different hospitals, offering a practical solution in scenarios where data availability is a major constraint. As we show in Table 4 and Table 5, to achieve the comparable results you will need to collect 25 times less data.
>
> **Grad-CAM and Its Modifications:** We agree that many methods could always be considered. However, we chose to focus on the baseline implementations of Grad-CAM to get an unbiased view of its performance in this context. Our method is not mutually exclusive from other ones. Anyone reading this paper can pick and choose whichever specific Grad-CAM method they want to use and adapt it to our proposed loss function. All you need to do is to update Equation 2 depending on a specific implementation of Grad-CAM. Everything else will be the same.
>
> **Diversity of Data:** In response to your feedback, Appendix C now includes comprehensive details on patient demographics, imaging equipment variations, and dataset distribution, reinforcing the robustness of our model against data shifts.
>
> **Class Imbalance:** We appreciate your pointing out the significance of class imbalance. For all experiments, we provide both AUROC and AUPRC to provide a balanced view of our model's performance, accounting for class distribution.
>
> **Literature Review and Future Work:** We have updated the literature review and introduced Appendix D with Future Work.

---

### Official Review · Reviewer_4V4q · 2024-02-28

**Confidence:** 4
**Preliminary Rating:** 3
**Final Rating:** 3.5

**Summary:**

The paper presents a method for improving medical image classification using segmentation masks and is particularly beneficial in scenarios with small datasets. This approach significantly improves model accuracy in diagnosing hydronephrosis by guiding the model's focus toward relevant features, showing substantial performance enhancements both on in-distribution (i.i.d) and out-of-distribution (OOD) datasets. The results demonstrate that models trained on smaller datasets using this method can match or outperform those trained on much larger datasets, emphasising the potential of segmentation-guided attention mechanisms in medical image analysis.

**Strengths:**

* Novelty: The paper introduces a novel method that integrates segmentation masks into the training process of medical image classification models.
* Improved performance on limited data: One of the significant strengths is the demonstration of the model's ability to achieve high accuracy with smaller datasets.
* Generalization capability: The paper shows that the proposed method improves model performance not just on in-distribution data but also on out-of-distribution datasets.
* Well-written paper: The paper adequately reviews prior work and contextualizes previous contributions. The methodology is rigorously explained.

**Weaknesses:**

* One of the weaknesses of the method is its dependency on segmentation masks which limits the practicality of the approach. In reality these masks may not be readily available or may be time-consuming and costly to produce.
* The paper lacks comparison with other state-of-the-art approaches. Experiments of the actual method are exhaustive and complete (even the hyperparameter selection is provided) but benchmarks with other methods are lacking making it hard to evaluate the actual improvement offered by the new approach.
* The dataset used is private, limiting reproducibility. Also, the data is not diverse in terms of demographics which might weaken the authors’ argument that the method works well on OOD datasets. No details were provided in terms of settings or imaging equipment used when collecting data.

**Detailed Comments:**

* Provide a more in-depth analysis of the data used. Since it is a private dataset, try to include patient demographics, disease prevalence, imaging equipment used or any other detail that shows OOD.
* Typos:
    - “The Domain Allignment” -> “Alignment”
    - “Self-supervised learning [...] let a model” -> “lets a model”

**Justification Of Final Rating:**

The authors have addressed my comments on the rebuttal process and I would like to thank them for their efforts. The paper introduces a method for training image classification models by guiding the model’s attention with segmentation masks. The approach is particularly efficient in small dataset scenarios. While the paper would benefit from additional experimental validations - especially to support the claims of DG-, the presented material is satisfactory. The reasons to accept this paper slightly outweigh the reasons to reject it.

**Justification Of The Preliminary Rating:**

The paper is well written and the provided approach is novel with enough experiments. However, without benchmarking against other state-of-the-art DG methods, it is challenging to gauge the actual progress provided by the method.

**Questions To Address In The Rebuttal:**

Providing a fair and thorough comparison with at least one other DG method.

---

> ### Author Response · Authors · 2024-03-18
>
> Thank you for your insightful comments and the opportunity to enhance our manuscript. Below we address the key points raised in your review:
>
> **Dependency on Segmentation Masks:** We appreciate your concern regarding the practicality of requiring segmentation masks. However, in specific medical conditions such as Hydronephrosis, creating segmentation masks for kidneys is relatively straightforward and cost-effective compared to the significant effort and resources needed to collect larger datasets. Our approach leverages these masks to train models that can be easily adapted and reused across different hospitals, offering a practical solution in scenarios where data availability is a major constraint. As we show in Table 4 and Table 5, to achieve comparable results you will need to collect 25 times less data.
>
> **Diversity of Data:** The limitation regarding data diversity is acknowledged in our manuscript. While our current datasets are primarily from Canada and the USA, we demonstrated our model's effectiveness (Table 3 and Table 5) across different hospitals and imaging equipment within these regions. We've added additional information about the datasets in Appendix C, highlighting variations in patient demographics and imaging equipment to confirm our claims about Out-of-Distribution performance. We’ve also added Appendix D about future work in which we will focus on extending this validation to datasets from other countries.
>
> **Future Work:** Our initial study was primarily designed to explore and validate the effectiveness of our segmentation-guided attention mechanism in scenarios characterized by small datasets and domain shifts — a niche that, to our understanding, had not been thoroughly investigated previously. Given this focus, and the complexities involved in directly comparing our method with a broad range of state-of-the-art domain generalization methods — each with its unique assumptions, requirements, and implementation details — we chose to concentrate our efforts on demonstrating the significant contributions our approach offers within its intended context. Nevertheless, your point is very important and we plan to focus on answering it in future work.
>
> **Typos:** Thank you for the careful read-through, we have fixed the typos.

---

### Official Review · Reviewer_T72m · 2024-02-28

**Confidence:** 4
**Preliminary Rating:** 3
**Recommendation:** Poster
**Final Rating:** 4

**Summary:**

This is a research paper that discusses the challenges of training machine learning models to analyze medical images. The paper proposes a new approach that uses attention maps to help models focus on the relevant parts of an image, such as a specific organ, while ignoring irrelevant background noise. The paper also introduces a new metric called Attention Score to measure how well the model is paying attention to the relevant areas. The paper highlights the importance of addressing the problem of domain shift, where models trained on one dataset may not perform well on another dataset. The proposed approach is shown to be effective on small datasets with less than 100 images.

**Strengths:**

The paper proposes a new approach for training medical image classification models using segmentation masks, which is particularly effective in small dataset scenarios. The authors claim that their approach significantly improves accuracy for diagnosing Hydronephrosis by guiding the model's attention with segmentation masks towards relevant features.

The paper also provides a comprehensive evaluation of the proposed model on identically distributed data, showing either the same or better performance with improvement up to 0.28 in AUROC. Besides, the authors introduce a new metric, Attention Score, to quantify how much different models pay attention to the region of interest, which is a valuable contribution to the field.

In terms of scientific merits, the paper follows general scientific principles and adequately addresses prior work. The authors also conducted a hyperparameter search to identify optimal hyperparameters for training all models, which makes the paper more scientifically rigorous.

**Weaknesses:**

This paper lacks novelty. It adopts Gradientweighted Class Activation Mapping (Grad-CAM) (Selvaraju et al., 2017) to compute an attention map. Also, their training set is not randomly selected, but the ones with segmentation masks. Thus, the fair comparison of their method to other methods or models is questionable.

**Detailed Comments:**

The paper could benefit from a more detailed explanation of the segmentation-guided attention approach. Possible explanations can be as follows:
If the segmentation automatic is fully automatic?
What segmentation method is used?

Ablation study for their processes might support this paper better.

**Justification Of Final Rating:**

This is a research paper that discusses the challenges of training machine learning models to analyze medical images. The paper proposes a new approach that uses attention maps to help models focus on the relevant parts of an image, such as a specific organ, while ignoring irrelevant background noise. The paper also introduces a new metric called Attention Score to measure how well the model is paying attention to the relevant areas. The paper highlights the importance of addressing the problem of domain shift, where models trained on one dataset may not perform well on another dataset. The proposed approach is shown to be effective on small datasets with less than 100 images.

From the following comments of the authors, now I understand better, resulting in change of my intial evaluation on this paper one step higher:

The segmentation masks are manually created by medical professionals our innovation
Its innovation lies in integrating these maps with a novel loss function

**Justification Of The Preliminary Rating:**

This paper lacks novelty. It adopts Gradient weighted Class Activation Mapping (Grad-CAM) to compute an attention map. Also, their training set is not randomly selected, but the ones with segmentation masks. Thus, the fair comparison of their method to other methods or models is questionable. Ablation study may strengthen this paper.

**Questions To Address In The Rebuttal:**

If the segmentation guided approach is applied to the other models in the experiments, this paper might be considered as a more acceptable paper.

**Special Issue:**

No

---

> ### Author Response · Authors · 2024-03-18
>
> We greatly appreciate your insights and the opportunity to clarify aspects of our work. Below, we address each of your concerns:
>
> **Grad-CAM and Novelty:** We want to flag a potential misunderstanding here. We do use Grad-CAM to compute the attention maps but our innovation lies in integrating these maps with a novel loss function. This function penalizes the model for focusing on irrelevant regions, improving accuracy in diagnosing Hydronephrosis, especially in small datasets — a key novelty of our approach.
>
> **Training Dataset Selection:** Our methodology includes a fair comparison through two baseline settings: 1) Models trained on images with segmentation masks (same as our model), and 2) Models trained on a dataset 25 times larger. This dual approach ensures a balanced evaluation of our model's effectiveness cause in the first case all models are trained on the same amount of data and the same images and in the second case only baseline models are trained on a bigger dataset.
>
> **Segmentation Masks Origin:** We want to highlight another potential misunderstanding. The segmentation masks are manually created by medical professionals, underscoring the practical and real-world applicability of our method in leveraging existing clinical annotations for model improvement.
>
> **Ablation Study:** The comparison between ResNet-18 models using Binary Cross Entropy loss versus our Attention loss (Tables 2 and 3) serves as an ablation study. This demonstrates significant improvements with our approach, both in identically distributed data and, notably, in OOD scenarios—highlighting the effectiveness of our loss function.
>
> **Future Work:** Recognizing the value of applying our segmentation-guided approach to other models, we aim to explore this in future research. Limited by computational resources, we couldn't include these experiments within the rebuttal period but have noted this in Appendix D in our revised manuscript.
>
> We believe these clarifications and our manuscript's revisions directly address your concerns, further underscoring our contributions to advancing medical image classification.

---

> > ### Comment · Reviewer_T72m · 2024-03-27
> > **increase my evaluation by one step higher**
> >
> > From the following comments of the authors, now I understand better, resulting in change of my intial evaluation on this paper one step higher:
> >
> > * The segmentation masks are manually created by medical professionals our innovation
> > * Its innovation lies in integrating these maps with a novel loss function

---

### Author Response · Authors · 2024-03-18

We would like to thank everyone for their time and feedback! We are encouraged by the positive remarks on the significance of our work in studying identically distributed and out-of-distribution medical image classification in small dataset scenarios.

We have responded to each reviewer individually, and following the feedback, we have made the following improvements:

1. Updated the introduction section and changed the title to better reflect the scope of our work
2. Added Appendix C with information about the datasets to showcase various data distribution
3. Included Appendix D, in which we discuss future work

We note the favorable comments from the reviewers, indicating that our approach is novel [T72m, 4V4q], the evaluation is comprehensive [T72m], and the methodology is rigorously explained [4V4q]. Thank you also for highlighting that attention Score is a valuable contribution to the field [T72m], our method could be useful for other ML/DL applications [HeNo], and the problem we solve is very important in medical image analysis [xY5m].

We thank everyone for their feedback and engagement and are ready to answer any further questions.

---

### Comment · Area_Chair_vHMW · 2024-03-18
**Please read and respond to author comments**

Dear reviewers. The authors have posted responses to your reviews. Please take the time to read and respond before March 27.

---

### Meta-Review · Area_Chair_vHMW · 2024-04-04

**Recommendation:** Accept (Poster)
**Confidence:** 4

**Metareview:**

This paper introduces a loss function between the attention of a model computed by GradCAM and segmentation masks to improve image classification performance in small dataset scenarios. The idea was in general found interesting by the reviewer, but there were some concerns about novelty, comparisons to other SOTA approaches, difficulty in generation of segmentation masks. Novelty concerns were resolved in the rebuttal phase though concerns around limited comparisons and difficulty in obtaining segmentation masks remained to some degree. Overall, I agree with the reviewers that this is a borderline paper with positives slightly outweighing the negatives.

---

### Decision · Program_Chairs · 2024-04-05

Accept (Poster)